# Compositing effects for high thermoelectric performance of Cu₂Se-based materials

Zhifang Zhou[1], Yi Huang[2,3], Bin Wei[1,4], Yueyang Yang[1], Dehong Yu[5], Yunpeng Zheng[1], Dongsheng He[6], Wenyu Zhang[1], Mingchu Zou[1], Jin-Le Lan[7]✉, Jiaqing He[2], Ce-Wen Nan[1] & Yuan-Hua Lin[1]✉

Thermoelectric materials can realize direct conversion between heat and electricity, showing excellent potential for waste heat recovery. Cu₂Se is a typical superionic conductor thermoelectric material having extraordinary $ZT$ values, but its superionic feature causes poor service stability and low mobility. Here, we reported a fast preparation method of self-propagating high-temperature synthesis to realize in situ compositing of BiCuSeO and Cu₂Se to optimize the service stability. Additionally, using the interface design by introducing graphene in these composites, the carrier mobility could be obviously enhanced, and the strong phonon scatterings could lead to lower lattice thermal conductivity. Ultimately, the Cu₂Se-BiCuSeO-graphene composites presented excellent thermoelectric properties with a $ZT_{max}$ value of ~2.82 at 1000 K and a $ZT_{ave}$ value of ~1.73 from 473 K to 1000 K. This work provides a facile and effective strategy to largely improve the performance of Cu₂Se-based thermoelectric materials, which could be further adopted in other thermoelectric systems.

Waste heat will be a potential and huge energy resource if it can be properly utilized. Based on carrier and phonon transport, thermoelectric technology with the capacity of direct energy conversion between heat and electricity could be an attractive alternative for waste heat recovery, which has been widely investigated in the past decades[1–4]. A good thermoelectric performance is needed for further application, which is generally determined by the dimensionless figure of merit, $ZT = S^2\sigma T/\kappa$. The parameters $S$, $\sigma$, $\kappa$, and $T$ stand for the Seebeck coefficient, electrical conductivity, thermal conductivity, and absolute temperature, respectively. Thus, achieving an outstanding $ZT$ value relies on a large power factor (PF = $S^2\sigma$) and a low $\kappa$, where the $\kappa$ is mainly consisted of carrier thermal conductivity ($\kappa_c$) and lattice thermal conductivity ($\kappa_l$)[5,6]. However, the strongly coupled correlation among $S$,

$\sigma$, and $\kappa_c$ has always been a big challenge for improving $ZT$ values[7,8]. To overcome this problem, various strategies have been proposed, such as band structure engineering[9–11], mobility optimization[12–14], lattice thermal conductivity suppression[15–17], and so on. To date, polycrystalline SnSe recorded the highest $ZT$ value of ~3.1 at 783 K[18]. The maximum $ZT$ ($ZT_{max}$) values over 2.0 have also been reported in Cu₂Se[19,20], PbTe[21,22], GeTe[23,24], and single-crystal SnSe[25,26] thermoelectric materials.

Due to the elemental abundance, non-toxicity, and wide working temperature range, Cu₂Se-based thermoelectric materials with high $ZT$ values are highly promising and notable in mass production and commercial applications[27–29]. The reason for the high thermoelectric performance of Cu₂Se could mainly be the highly mobile Cu⁺ ions, which cause strong phonon scattering and further lead to intrinsically

[1]State Key Laboratory of New Ceramics and Fine Processing, School of Materials Science and Engineering, Tsinghua University, 100084 Beijing, China. [2]Shenzhen Key Laboratory of Thermoelectric Materials, Department of Physics, Southern University of Science and Technology, 518055 Shenzhen, China. [3]Analytical Instrumentation Center, Hunan University, 410000 Changsha, China. [4]Henan Key Laboratory of Materials on Deep-Earth Engineering, School of Materials Science and Engineering, Henan Polytechnic University, 454000 Jiaozuo, China. [5]Australian Nuclear Science and Technology Organisation, Lucas Heights 2234 New South Wales, Australia. [6]Core Research Facilities, Southern University of Science and Technology, 518055 Shenzhen, China. [7]State Key Laboratory of Organic-Inorganic Composites, College of Materials Science and Engineering, Beijing University of Chemical Technology, 100029 Beijing, China. ✉e-mail: lanjl@mail.buct.edu.cn; linyh@mail.tsinghua.edu.cn

low $\kappa_l$[30]. Meanwhile, the superionic nature has detrimental effects on the stability of $Cu_2Se$ and then damages its thermoelectric performance under some certain application conditions[27,28,31]. Thus, improving the stability and remaining a high $ZT$ value is a challenge for $Cu_2Se$-based materials. Inhibiting the long-range migration of $Cu^+$ ions by introducing ion-blocking interfaces has been proposed as an effective strategy to improve the stability of $Cu_2Se$[32–34]. It is proposed that $Cu_2Se$/BiCuSeO interfaces could stabilize $Cu_2Se$ by blocking the long-range migration of $Cu^+$ ions and obstructing the reduction reaction of $Cu^+$ into Cu metal precipitates[34].

However, the relatively low carrier mobility of $Cu_2Se$ (lower than 20 cm$^2$ V$^{-1}$ s$^{-1}$) and BiCuSeO (lower than 10 cm$^2$ V$^{-1}$ s$^{-1}$) hinders the further improvement of thermoelectric performance, especially the average $ZT$ ($ZT_{ave}$) values[19,34–39]. Thus, optimizing carrier mobility is urgent for achieving higher $ZT$ values of $Cu_2Se$-based materials. Recently, carbon materials have been attractive in optimizing thermoelectric performance since it has excellent electrical properties[40–42], which can introduce extra electrically conductive paths and are expected for improving the carrier mobility of composites. In addition, previous work has shown that introducing carbon structures can also decrease lattice thermal conductivity because of the interfacial phonon scattering and strong phonon reflection at phase interfaces[20,38,41].

Considering the higher thermoelectric properties of $Bi_{0.88}Pb_{0.06}Ca_{0.06}CuSeO$ (BPCCSO)[35] than that of pristine BiCuSeO and the robust level of carrier mobility in graphene[40,41], the $Cu_2Se$-BPCCSO-graphene composites were designed and prepared by a fast method of combining self-propagating high-temperature synthesis (SHS) with spark plasma sintering (SPS). We found that compositing with BPCCSO and graphene synergistically optimized electrical and thermal properties. Excellent electrical performance with the highest average PF (PF$_{ave}$) value of 14.49 μW cm$^{-1}$ K$^{-2}$ from 473 K to 1000 K was obtained, which could be mainly attributed to the optimized weighted mobility. Meanwhile, the multiple interfaces brought strong phonon scatterings to achieve a relatively low lattice thermal conductivity value of 0.11 W m$^{-1}$ K$^{-1}$ at 1000 K. Ultimately, attractively high $ZT$ values including a $ZT_{max}$ value of ~2.82 at 1000 K and a $ZT_{ave}$ value of ~1.73 from 473 K to 1000 K were realized, which exhibited good potential for future practical applications.

## Results

### $Cu_2Se$-BPCCSO composites

By using the SHS-SPS method, we first prepared $Cu_2Se$-BPCCSO composites and explored the optimum compositions for further investigation. For comparison, $Cu_2Se$ and BPCCSO were also prepared by the same method in the meantime. All the samples were highly dense with a relative density of over 95%, as shown in Supplementary Table 1. By in situ compositing with BPCCSO, the electrical conductivity of $Cu_2Se$-BPCCSO composites in the whole temperature range was much higher than that of $Cu_2Se$ (Fig. 1a), which was mainly due to the improved carrier concentration by the composite of BPCCSO with large carrier concentration (4.39×10$^{20}$ cm$^{-3}$). For instance, the carrier concentration increased from 2.22×10$^{20}$ cm$^{-3}$ of $Cu_2Se$ to 2.80×10$^{20}$ cm$^{-3}$ of 0.9$Cu_2Se$-0.1BPCCSO (Supplementary Table 2). Although $Cu_2Se$-BPCCSO composites showed lower Seebeck coefficient values than that of $Cu_2Se$ (Fig. 1b), $Cu_2Se$-BPCCSO composites possessed higher PF values (Fig. 1c). Furthermore, we performed the thermal conductivity of the samples, as shown in Fig. 1d. The thermal conductivity was calculated by the equation, $\kappa=D\rho C_p$, where temperature-dependent thermal

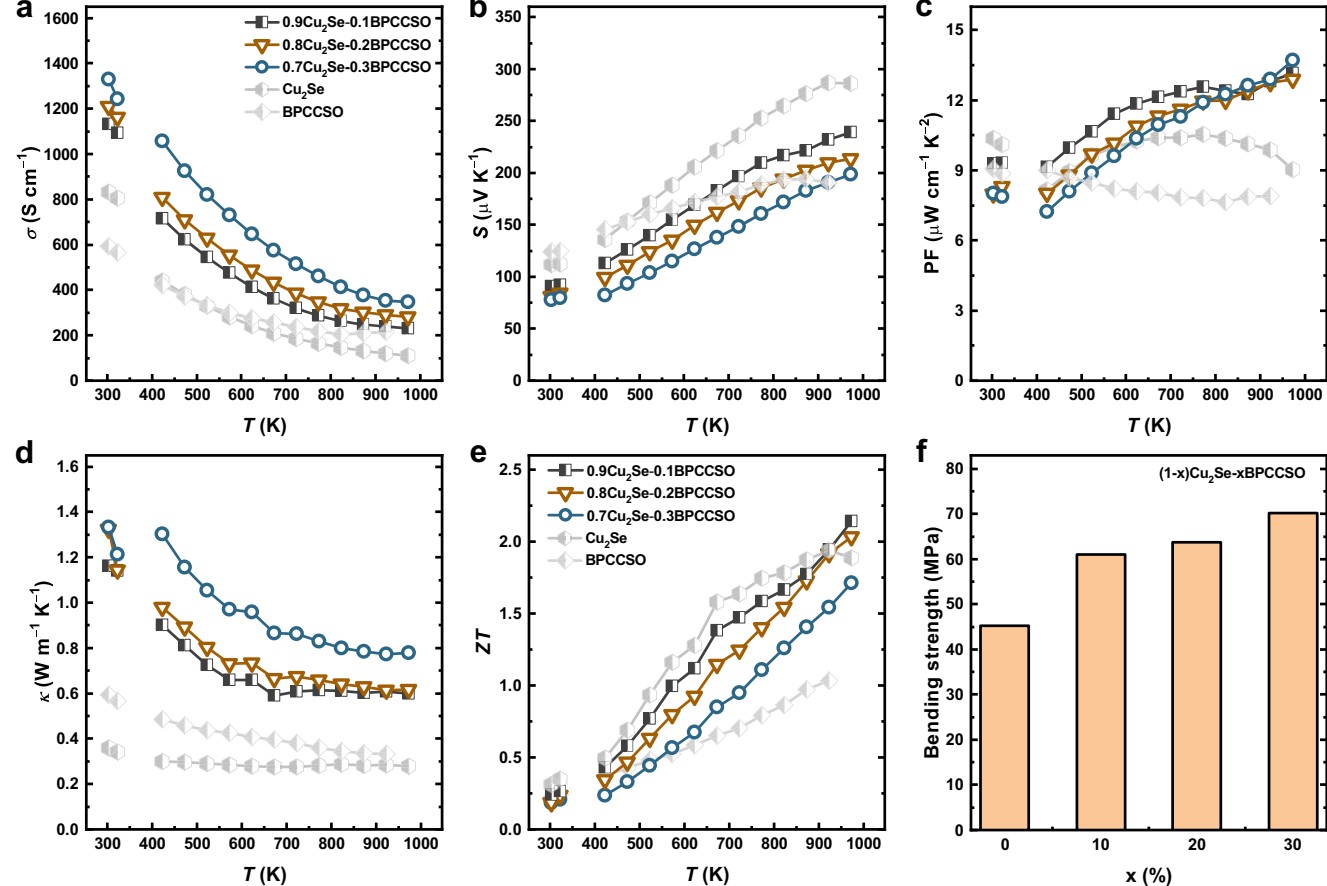

**Fig. 1 | Thermoelectric and mechanical properties of $Cu_2Se$-BPCCSO composites. a** Electrical conductivity ($\sigma$). **b** Seebeck coefficient ($S$). **c** Power factor (PF). **d** Thermal conductivity ($\kappa$). **e** $ZT$ values. **f** Bending strength.

diffusivity ($D$) is shown in Supplementary Fig.1a, density ($\rho$) can be seen in Supplementary Table 1, and specific heat capacity ($C_p$) was estimated according to the Neumann–Kopp rule based on the measured $C_p$ of Cu$_2$Se in this work (The measured $C_p$ of Cu$_2$Se is shown in Supplementary Fig. 2). Since the total thermal conductivity is mainly consisted of carrier thermal conductivity and lattice thermal conductivity, it can be seen that the samples with higher electrical conductivity possessed higher thermal conductivity. Besides, we calculated the carrier thermal conductivity according to the Wiedemann–Franz law, $\kappa_c = L\sigma T$, and then obtained the lattice thermal conductivity ($\kappa_l$). The temperature-dependent $\kappa_l$ and Lorentz constant ($L$) are presented in Supplementary Fig. 1b, c. Compared to Cu$_2$Se and BPCCSO, the Cu$_2$Se-BPCCSO composites had lower lattice thermal conductivity due to the interfacial phonon scatterings.

Combined the electrical and thermal properties, the $ZT$ values were calculated and shown in Fig. 1e, where the 0.9Cu$_2$Se-0.1BPCCSO sample possessed the highest $ZT_{max}$ value of ~2.14 at 973 K. To test the service stability of such composites, we re-measured the 0.9Cu$_2$Se-0.1BPCCSO sample and conducted heating and cooling cycles test. As shown in Supplementary Fig. 3, the thermoelectric properties including electrical conductivity, Seebeck coefficient, thermal conductivity, and $ZT$ values all presented excellent stability with only small changes under experimental uncertainty. Furthermore, it is proposed that electrical conductivity has a positive correlation with critical voltage and can be a simple indicator to rapidly predict the stability of Cu$_2$Se-based materials, where the higher electrical conductivity is, the better stability will be[43]. Therefore, the Cu$_2$Se-BPCCSO composites should have better stability than Cu$_2$Se. In addition, with the compositing of BPCCSO, the bending strength of Cu$_2$Se-BPCCSO composites largely improved from ~45 MPa to ~70 MPa (Fig. 1f). To conclude, by in situ compositing with BPCCSO, Cu$_2$Se-BPCCSO composites exhibited promising thermoelectric properties, excellent thermal stability, and good mechanical properties. However, although 0.9Cu$_2$Se-0.1BPCCSO showed the highest $ZT_{max}$ value, it is noted that the relatively low carrier mobility (Supplementary Table 2) and high lattice thermal conductivity still limited its thermoelectric properties for further applications. Considering the synergistic effects of graphene on electrical and thermal transport properties, Cu$_2$Se-BPCCSO-graphene composites were designed to obtain better thermoelectric performance.

## Structure of Cu$_2$Se-BPCCSO-graphene composites

The phase structure of 0.9Cu$_2$Se-0.1BPCCSO-$x$ wt% graphene composites ($x$ = 0, 0.01, 0.02, 0.03, 0.035, 0.04) can be analyzed via the X-ray diffraction (XRD) patterns, as shown in Supplementary Fig. 4. It was obvious that the diffraction patterns of 0.9Cu$_2$Se-0.1BPCCSO-$x$ wt% graphene composites ($x$ = 0, 0.01, 0.02, 0.03, 0.035, 0.04) were almost the same and mainly consisted of two phases, Cu$_2$Se and BPCCSO, indicating the high efficiency of the facile and fast preparation method for fabricating such complex composites. Owing to the low content of graphene, it was difficult to detect its phase by XRD. Besides the main phases, two minor peaks of Bi$_2$O$_3$ around 27.5° and 32.7° were observed in composites as well, which formed different scales of precipitates embedded in the matrix to strengthen phonon scatterings, bringing a further decrease in lattice thermal conductivity. The existence of Bi$_2$O$_3$ may be ascribed to the evaporation of Bi elements, and a small amount of Bi$_2$O$_3$ was unreacted as impurity phases.

The fracture microstructure of 0.9Cu$_2$Se-0.1BPCCSO-$x$ wt% graphene composites ($x$ = 0, 0.01, 0.02, 0.03, 0.035, 0.04) is presented in Supplementary Fig. 5. Two kinds of areas with different contrasts existed in each sample revealed by the scanning electron microscopy (SEM) images in backscattering electron imaging (BEI) mode. The areas with brighter contrast were BPCCSO, while the others were Cu$_2$Se, which was confirmed by energy dispersive spectroscopy (EDS) (Supplementary Fig. 6). To further investigate the interfaces between each phase, we conducted high-angle annular dark field (HAADF) analyses using scanning transmission electron microscopy (STEM). The two main phases including Cu$_2$Se and BPCCSO could be seen, as shown in Fig. 2a. In addition, a few nanometer-sized Bi$_2$O$_3$ secondary phases were also observed, which was consistent with the XRD results. Interestingly, the additional Cu-Se layer and Cu layer were coherently intercalated in BPCCSO lattice when composited with other phases (Fig. 2b, d, e), which was different from BPCCSO observed in other areas (Supplementary Fig. 7). Parts of high mobile Cu$^+$ could be trapped by such structure, which limited the long migration of Cu$^+$ and further improved the stability. Moreover, the Cu$_2$Se-BPCCSO interface was smooth and coherent in some specific directions, as shown in Fig. 2c. The parallel orientation relationships between the two phases could be reflected by the selected area electron diffraction (SAED) patterns (Fig. 2f, g) that were taken from BPCCSO and Cu$_2$Se. For instance, the (001) plane of BPCCSO was parallel to the (−206) plane of Cu$_2$Se. Such in situ formed interfaces can effectively scatter phonons without impeding carrier transport, which is beneficial to decoupling carrier-phonon interactions. In addition, graphene with different sizes has been found in composites, as shown in Supplementary Fig. 8, proving the existence of graphene.

## Charge transport properties

The temperature-dependent electrical conductivities of 0.9Cu$_2$Se-0.1BPCCSO-$x$ wt% graphene composites ($x$ = 0, 0.01, 0.02, 0.03, 0.035, 0.04) are presented in Fig. 3a. Almost all the samples behaved in the same trend that the electrical conductivity decreased with rising temperature, indicating degenerate semiconductors. With increasing the contents of graphene, the electrical conductivity first increased and then decreased. Compared to 0.9Cu$_2$Se-0.1BPCCSO composite, the addition of graphene ($x$ = 0.02, 0.03, 0.035) further increased the electrical conductivity of the Cu$_2$Se-BPCCSO-graphene composites, where the maximum value of 1538 S cm$^{-1}$ was 36% higher than that of the 0.9Cu$_2$Se-0.1BPCCSO composite without graphene and the value was approximately twice than that of Cu$_2$Se. The results could be attributed to that the appropriate content of graphene could introduce extra electrically conductive paths and effectively improve the mobility of composites. Here, the Hall mobility reached 39.63 cm$^2$ V$^{-1}$ s$^{-1}$, which was 57% higher than that of the 0.9Cu$_2$Se-0.1BPCCSO composite (Fig. 3b and Supplementary Table 2). It is also noted that when the content of graphene reached 0.04 wt%, the Hall mobility decreased sharply, which caused the lower electrical conductivity of 0.9Cu$_2$Se-0.1BPCCSO-0.04 wt% graphene composite. Meanwhile, we found that the addition of graphene had only a small effect on the carrier concentration and the carrier concentration of samples showed a slightly decreasing trend with increasing graphene content, which also indicated that the introduced graphene was n-type. Compared to the BPCCSO and Cu$_2$Se-based materials in the literature[19,34–39], as Fig. 3c shows, the composites in this work had a moderate carrier concentration and the highest Hall mobility, which was conducive to achieving better electrical transport performance.

The temperature-dependent Seebeck coefficient of 0.9Cu$_2$Se-0.1BPCCSO-$x$ wt% graphene composites ($x$ = 0, 0.01, 0.02, 0.03, 0.035, 0.04) is shown in Fig. 3d. Since the Seebeck coefficients of all samples were positive, the majority of carriers were holes. The change trend of the Seebeck coefficient for samples with different graphene contents was related to the electrical conductivity, but the variety was small, especially at room temperature, suggesting that the existence of graphene with appropriate contents decoupled the correlation between electrical conductivity and Seebeck coefficient to a certain extent. Based on the results of the Seebeck coefficient and electrical conductivity, the temperature-dependent PF values were calculated, as presented in Fig. 3e. Attributed to the much higher electrical conductivity, the PF values of the 0.9Cu$_2$Se-0.1BPCCSO-$x$ wt% graphene composites ($x$ = 0.02, 0.03, 0.035) were significantly improved within the entire temperature range, reaching 16.97 μW cm$^{-1}$ K$^{-2}$ at 1000 K of

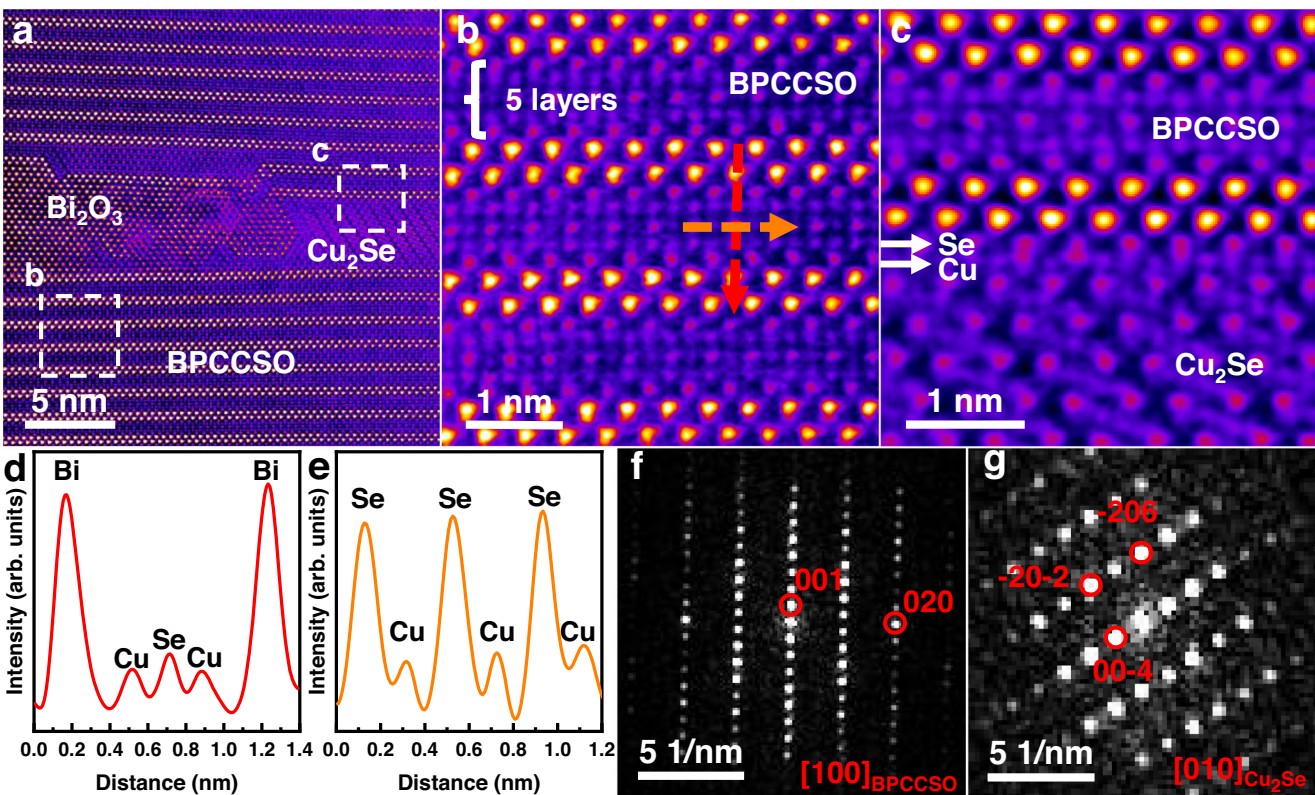

**Fig. 2 | Atomic-scale interfacial structure.** The HAADF-STEM images show **a** overview of the phase interfaces. **b, c** Enlarged areas (b, c) for BPCCSO and Cu$_2$Se-BPCCSO interface. **d, e** Intensity line profiles marked by the red and orange dashed arrows in **b. f, g** The diffractograms of BPCCSO and Cu$_2$Se, respectively, which show the orientation relationship of the two phases.

0.9Cu$_2$Se-0.1BPCCSO-0.035 wt% graphene, which was much higher than that of the Cu$_2$Se-BPCCSO composite and Cu$_2$Se sample. At the same time, the average PF values in the mid-to-high temperature range significantly increased. The average PF value of 0.9Cu$_2$Se-0.1BPCCSO-0.035 wt% graphene reached 14.49 μW cm$^{-1}$ K$^{-2}$ in the temperature range from 473 K to 1000 K (Supplementary Fig. 9). To our best knowledge, both the peak PF value and average PF value achieved here are the highest values compared to PF values from the current state-of-the-art Cu$_2$Se-based thermoelectric materials in the literature[19,20,34,36,44,45], indicating the excellent electrical performance of the Cu$_2$Se-BPCCSO-graphene composites in this work.

To get a better understanding of inherent electrical transport properties, the weighted mobility of the composites was calculated based on the measured Seebeck coefficient and electrical conductivity, as shown in Fig. 3f, and further details can be found in the Supplementary text or elsewhere[46,47]. The weighted mobility of all samples showed a $T^{-3/2}$ correlation with temperature, indicating that acoustic phonon scattering was the main carrier scattering mechanism[46]. It is also noted that the 0.9Cu$_2$Se-0.1BPCCSO-$x$ wt% graphene ($x$ = 0.02, 0.03, 0.035) composites possessed much higher weighted mobility than others, which was consistent with the Hall mobility and PF values, proving their good electrical transport properties.

**Thermal transport properties**
In addition to the excellent electrical transport properties, Cu$_2$Se-based thermoelectric materials are well known for their relatively low thermal conductivity due to their superionic nature. The temperature-dependent thermal conductivity of 0.9Cu$_2$Se-0.1BPCCSO-$x$ wt% graphene composites ($x$ = 0, 0.01, 0.02, 0.03, 0.035, 0.04) is shown in Fig. 4a. The related thermal diffusivity is shown in Supplementary Fig. 10. With the increasing contents of

graphene from $x$ = 0.02 to $x$ = 0.04, the thermal conductivity first increased and then decreased. Compared to 0.9Cu$_2$Se-0.1BPCCSO, although 0.9Cu$_2$Se-0.1BPCCSO-0.035 wt% graphene showed higher electrical conductivity, they had close thermal conductivity, which may be due to the much lower lattice thermal conductivity. Therefore, by subtracting carrier thermal conductivity, we obtained the lattice thermal conductivity of 0.9Cu$_2$Se-0.1BPCCSO-$x$ wt% graphene composites ($x$ = 0, 0.01, 0.02, 0.03, 0.035, 0.04). The temperature-dependent lattice thermal conductivity and Lorentz constant ($L$) are presented in Supplementary Fig. 11a, b. Although 0.9Cu$_2$Se-0.1BPCCSO composite sample possessed a relatively low lattice thermal conductivity, it is noted that the lattice thermal conductivity of the 0.9Cu$_2$Se-0.1BPCCSO-$x$ wt% graphene ($x$ = 0.01, 0.02, 0.03, 0.035, 0.04) composite samples was even lower. With the increasing content of graphene, the lattice thermal conductivity increased, which may be ascribed to the aggregation of graphene that reduced the interfacial phonon scattering effects. As revealed by the STEM results, due to the increase of multiple interfaces among BPCCSO, Cu$_2$Se, and graphene, more phonon scattering sources strengthened the phonon scattering and improved the interfacial thermal resistance.

Furthermore, by measuring sound velocity, the longitudinal sound velocity ($v_l$) and shear sound velocity ($v_s$) were obtained. And the average sound velocity ($v_a$), phonon mean free path ($l_{ph}$), and elastic properties including Young's modulus ($E$), Poisson ratio ($v_p$), and Grüneisen parameter ($\gamma$) were calculated and listed in Supplementary Table 3. The detailed calculation formulas are provided in the Supplementary text. The $v_a$ and $l_{ph}$ values of 0.9Cu$_2$Se-0.1BPCCSO-$x$ wt % graphene ($x$ = 0, 0.01, 0.02, 0.03, 0.035, 0.04) were also plotted in Fig. 4b. It is found that the addition of graphene would significantly reduce the sound velocity of the samples from 1585 m s$^{-1}$ to 1428 m s$^{-1}$, which meant the softening of the phonons. Meanwhile, the declined

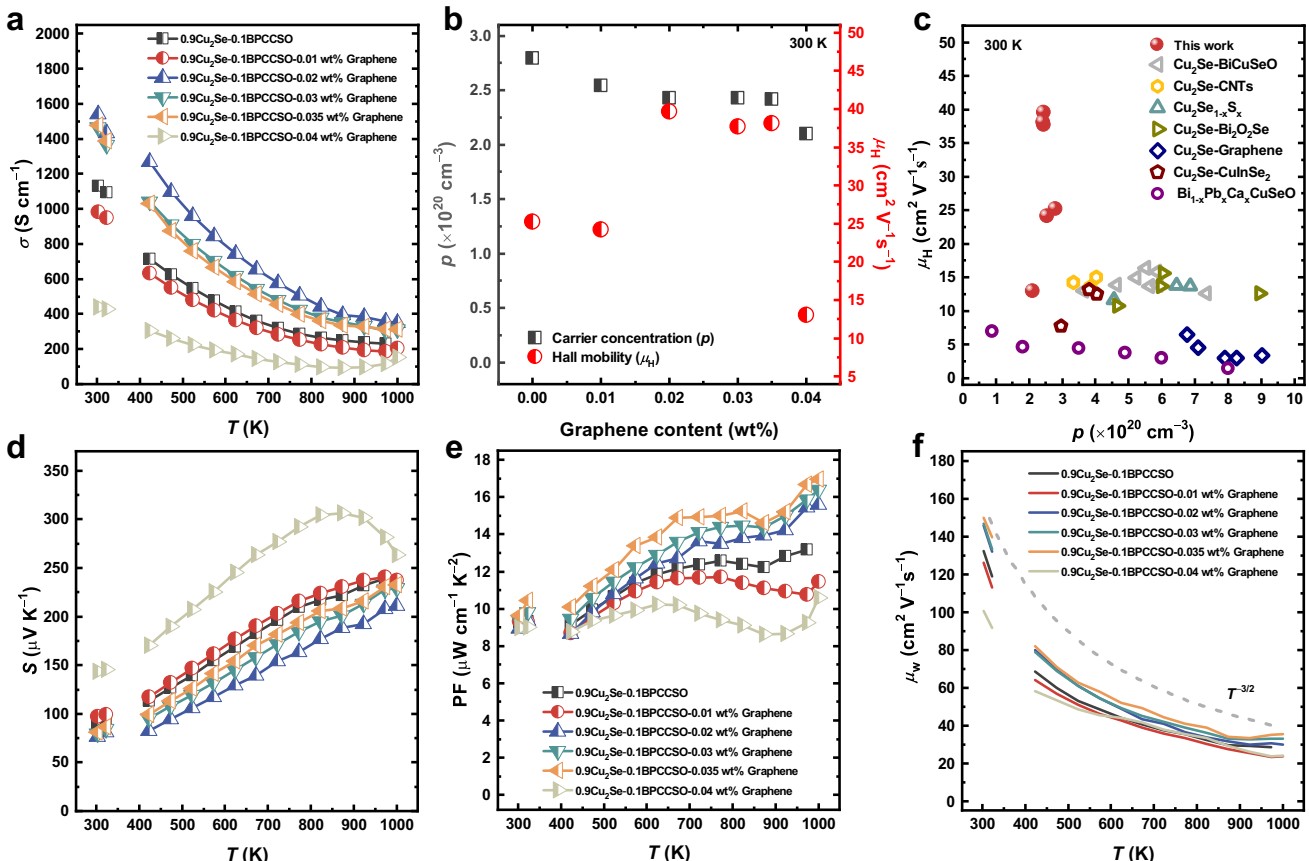

**Fig. 3 | Charge transport properties. a** Electrical conductivity ($\sigma$). **b** Room temperature Hall measurement results of 0.9Cu$_2$Se-0.1BPCCSO-$x$ wt% graphene composites ($x$ = 0, 0.01, 0.02, 0.03, 0.035, 0.04), including carrier concentration ($p$) and Hall mobility ($\mu_H$). Hall measurement results of Cu$_2$Se and BPCCSO can be found in Supplementary Table 2. **c** A collection plot showing the carrier concentration ($p$) and Hall mobility ($\mu_H$) for 0.9Cu$_2$Se-0.1BPCCSO-$x$ wt% graphene composites ($x$ = 0, 0.01, 0.02, 0.03, 0.035, 0.04) for this study and other Cu$_2$Se-based thermoelectric materials as well as BPCCSO from the literature[19,34–39]. **d** Seebeck coefficient ($S$). **e** Power factor (PF). **f** Weighted mobility ($\mu_w$). The dashed line indicates $T^{-3/2}$ correlation.

phonon mean free path from 0.34 nm to 0.20 nm also proved the increase of phonon scattering.

To get deep insight into phonon transport properties, inelastic neutron scattering (INS) experiments were performed to have measured the generalized phonon density of states (GDOS) for Cu$_2$Se, BPCCSO, 0.9Cu$_2$Se-0.1BPCCSO, and 0.9Cu$_2$Se-0.1BPCCSO-0.035 wt% graphene. Figure 4c shows the experimental GDOS of Cu$_2$Se, BPCCSO, and nano graphite which was taken from the reference[48]. The majority of phonon modes for Cu$_2$Se and BPCCSO were present below 40 meV, while the phonon modes of nano graphite were mainly located far above 40 meV. Thus, the overlaps among Cu$_2$Se, BPCCSO, and nano graphite should be poor, indicating a weak interface thermal transport[49].

To quantitatively describe the overlapping of phonon modes, the overlapping factor ($\delta$) was introduced, which is defined as $\delta = \int H(\omega)d\omega$. The $H(\omega)$ is the normalized intersection height of GDOS at a frequency $\omega$[50,51]. The $\delta$ of Cu$_2$Se-BPCCSO-nano graphite was 0.376. As a comparison, the $\delta$ of Cu$_2$Se-nanographite, and BPCCSO-nano graphite were also calculated and the values were 0.437 and 0.514, respectively. The poor overlap provided a good explanation for the ultralow lattice thermal conductivity of Cu$_2$Se-BPCCSO-graphene composites. Moreover, the peaks shifted to lower energy when measured at 600 K (Fig. 4d) indicating the phonon softening as a result of anharmonic lattice dynamics. This will further decrease the lattice thermal conductivity at higher temperatures due to the enhanced phonon scattering. It is noted that while the phonon modes of graphene had not been clearly detected for the sample 0.9Cu$_2$Se-0.1BPCCSO-0.035 wt% graphene, as indicated in Fig. 4d, the graphene modes were indeed identified for composites with more graphene (Supplementary

Fig. 12). To conclude, the extremely low lattice thermal conductivity originated from the increase of phonon scattering and softening of the phonons.

## Thermoelectric figure of merit

The temperature-dependent $ZT$ values of 0.9Cu$_2$Se-0.1BPCCSO-$x$ wt% graphene composites ($x$ = 0, 0.01, 0.02, 0.03, 0.035, 0.04) were calculated and demonstrated in Fig. 5a. Combined the excellent electrical properties with low thermal conductivity, the $ZT$ values of the Cu$_2$Se-BPCCSO-graphene composites were significantly improved. The $ZT$ values of 0.9Cu$_2$Se-0.1BPCCSO-0.035 wt% graphene reached 2.82 at 1000 K. Compared to Cu$_2$Se and 0.9Cu$_2$Se-0.1BPCCSO, the $ZT_{max}$ value increased by ~50% and ~32%. Furthermore, the average $ZT$ ($ZT_{ave}$) values of 0.9Cu$_2$Se-0.1BPCCSO-0.035 wt% graphene in the temperature range from 473 K to 1000 K was over 1.7, which was superior to most Cu$_2$Se-based materials with good thermoelectric performance[19,20,34,36,44,45], as shown in Fig. 5b. Since an excellent thermoelectric material should simultaneously have high $ZT_{max}$ and $ZT_{ave}$ values, the Cu$_2$Se-BPCCSO-graphene composites in this work could meet the needs of thermoelectric devices.

## Discussion

In this study, Cu$_2$Se-BPCCSO-graphene composites were successfully prepared by a fast preparation method of combining self-propagating high-temperature synthesis (SHS) with spark plasma sintering (SPS). The Cu$_2$Se-BPCCSO-graphene composites presented excellent thermoelectric properties with a $ZT_{max}$ > 2.8 at 1000 K and $ZT_{ave}$ > 1.7 in the mid-to-high temperature range. The

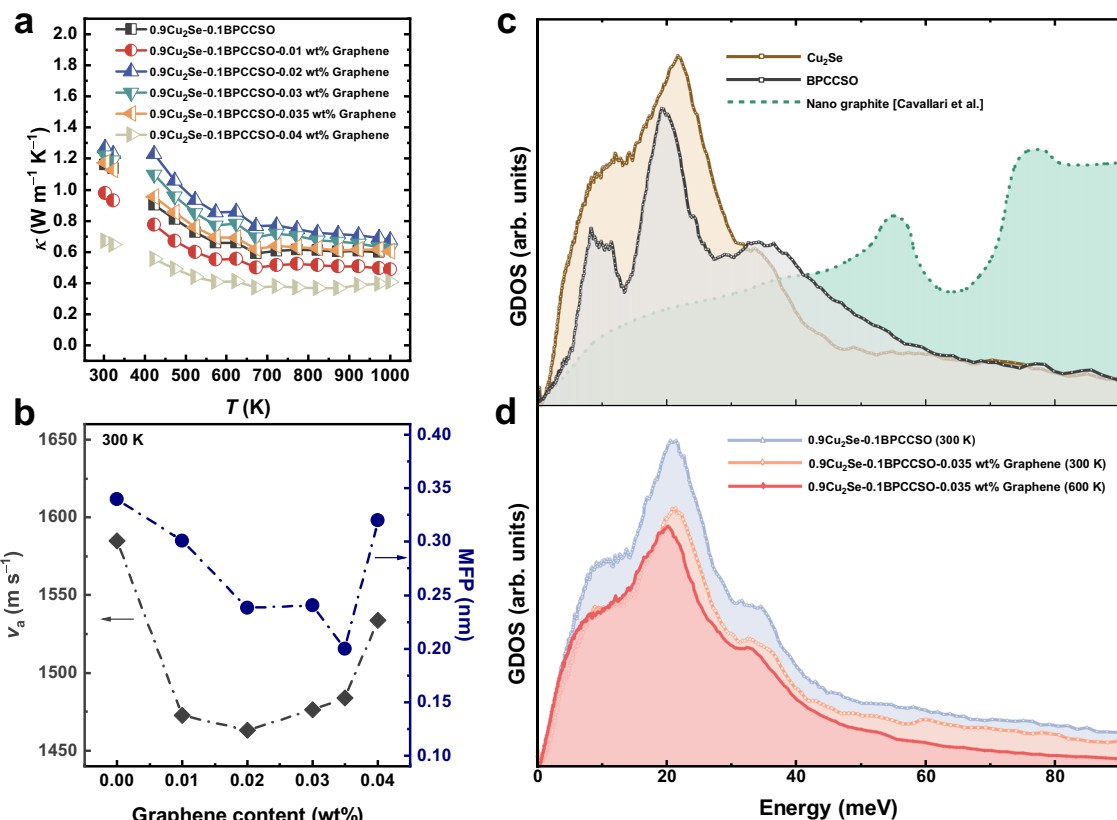

**Fig. 4 | Thermal transport properties. a** Total thermal conductivity ($\kappa$). **b** Room temperature average sound velocity ($v_a$) and phonon mean free path ($l_{ph}$). **c** The generalized phonon density of states (GDOS) of $Cu_2Se$, BPCCSO, measured via neutron spectroscopy at 300 K. Experimental GDOS of nano graphite was taken from reference[48]. **d** The experimental GDOS of 0.9$Cu_2$Se-0.1BPCCSO (300 K), 0.9$Cu_2$Se-0.1BPCCSO-0.035 wt% graphene (300 K and 600 K).

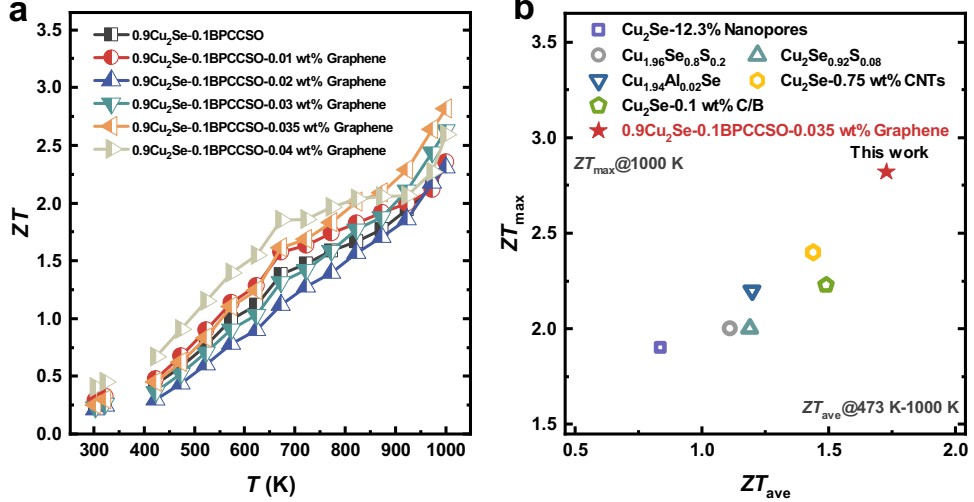

**Fig. 5 | Dimensionless figure of merit. a** Temperature-dependent $ZT$ values of 0.9$Cu_2$Se-0.1BPCCSO-$x$ wt% graphene composites ($x$ = 0, 0.01, 0.02, 0.03, 0.035, 0.04), BPCCSO and $Cu_2$Se. **b** Comparison of average $ZT$ values ($ZT_{ave}$ at 473–1000 K) and maximum $ZT$ values ($ZT_{max}$ at 1000 K) of 0.9$Cu_2$Se-0.1BPCCSO-0.035 wt% graphene composite in this work and other state-of-the-art $Cu_2$Se-based systems[19,20,34,36,44,45].

underlying mechanisms of carrier and phonon transport were deeply investigated. On one hand, the electrical performance was largely improved with the highest $PF_{max}$ value of 16.97 μW cm$^{-1}$ K$^{-2}$ at 1000 K and $PF_{ave}$ value of 14.49 μW cm$^{-1}$ K$^{-2}$ from 473 K to 1000 K, which could be attributed to the optimized carrier concentration and Hall mobility by compositing with BPCCSO and graphene. On the other hand, multiple scattering sources like interfaces between

BPCCSO, $Cu_2$Se, and graphene revealed by STEM strongly scattered phonons, resulting in extremely low lattice thermal conductivity. The weak interface thermal transport in such composites was further proved by inelastic neutron scattering (INS) techniques. This work gives a paradigm to improve the thermoelectric performance of $Cu_2$Se-like superionic semiconductors, which could be further adopted in other thermoelectric systems.

## Methods

### Sample preparation

The $0.9Cu_2Se-0.1Bi_{0.88}Pb_{0.06}Ca_{0.06}CuSeO$ (BPCCSO)-$x$ wt% graphene ($x = 0$, 0.01, 0.02, 0.03, 0.035, 0.04) composites were fabricated by a fast preparation method of combining self-propagating high-temperature synthesis (SHS) with spark plasma sintering (SPS). Stoichiometric amounts of Bi (99.99%, Innochem), $Bi_2O_3$ (99.9%, Meryer), Cu (AR, Meryer), Se (99.99%, Meryer), PbO (99%, Meryer), and CaO (99.99%, Aladdin) powders were thoroughly mixed by hand grinding, then cold pressed into pellets and underwent the SHS process. For each sample, the weight of the pellet was about 10 g. The pellet was put in an alumina crucible. The SHS process started by heating the bottom of the crucible to the ignition temperature with an alcohol lamp. Once the reaction began from the lower melting point component, we put the lid on the crucible and moved the alcohol lamp away. The combustion wave was persisted by the energy released from the initial reaction and spread to the whole pellet in few seconds. The obtained bulks were ground into fine powders and fully mixed with graphene (XFNANO, 0.5-5 μm in diameter, 0.8 nm in thickness) by hand grinding for more than 30 min, and then sintered by an SPS furnace at 973 K under a uniaxial pressure of 40 MPa for 3 min in vacuum.

### Structural characterization

The identification of phase purity and crystal structure for the composites was analyzed by X-ray diffraction (XRD, Bruker D8 Advance, Germany) with a Cu Kα radiation (λ = 0.15406 nm) at the scanning rate of 1° min⁻¹. The samples for XRD measurement were fine powders crushed and ground from the sintered bulks. The microstructure of different samples was observed by field-emission scanning electron microscopy (FESEM, MERLIN Compact, Car Zeiss, Germany) using backscattering electron imaging (BEI) mode at 15 kV. The crystallography, compositions, and interfacial structure were further investigated by a double $C_S$-corrected scanning transmission electron microscopy (STEM, Titan Themis Z, Thermo Fisher Scientific, the U. S.) equipped with probe and image correctors simultaneously. The probe convergence angle and high angle annular dark-field (HAADF) acceptance angles were 25 mrad and 48-200 mrad, respectively. The TEM specimens were prepared by ion milling with an Ar gas source, and the energy was 2 kV for perforation and 0.5 kV for the removal of the amorphous layer.

### Inelastic neutron spectroscopy (INS) measurements

The INS experiments were conducted on the time-of-flight cold neutron spectrometer, Pelican, at the Australian Nuclear Science and Technology Organisation (ANSTO). The five powder samples are $Cu_2Se$, BPCCSO, $0.9Cu_2Se-0.1BPCCSO$, $0.9Cu_2Se-0.1BPCCSO-0.035$ wt% graphene, and $0.9Cu_2Se-0.1BPCCSO-0.2$ wt% graphene. The measurement temperatures were 300 K and 600 K. The wavelength of the incident neutrons was aligned for 4.75 Å, corresponding to an energy of 3.63 meV. Background subtraction of the empty sample can and normalization to a vanadium sample, used as the isotropic scatterer, for correcting detector efficiency were also carried out. Finally, the scattering function was converted to the generalized phonon density of states (GDOS) as a function of energy transfer at the neutron-energy-gain side. The GDOS accounts for the temperature dependence of the scattering function, which is related to the thermal population of excitations.

### Transport property measurement

The values of electrical conductivity ($\sigma$) and Seebeck coefficient ($S$) were measured by a commercial thermoelectric measurement system (ZEM-3, ULVAC-RIKO, Japan) from room temperature to 1000 K under the protective atmosphere of helium gas. The sample size was about $3 \times 3 \times 11$ mm³. The carrier concentration ($n$) and Hall mobility ($\mu_H$) were determined from room temperature Hall coefficient ($R_H$) measurements by using the van der Pauw method on a homemade system

equipped with a maximum 5 $T$ superconducting magnet (Cryogenic Limited, U.K.). The sample size was about $10 \times 10 \times 0.5$ mm³. Thermal conductivity ($\kappa$) was calculated from the equation, $\kappa = D\rho C_p$, where $\rho$ is the density of the specimen, $C_p$ is the specific heat capacity, and $D$ is the thermal diffusivity, respectively. The $\rho$ was measured by the Archimedes method. The $C_p$ was measured by differential scanning calorimetry on a DSC (STA 409PC/DIL 402 PC, Netzsch, Germany) under an argon atmosphere and estimated according to the Neumann–Kopp rule in this work. And the $D$ was measured by a laser flash method (LFA 457, Netzsch, Germany) from room temperature to 1000 K under a continuous argon flow. The longitudinal and shear sound velocities were obtained at room temperature via an ultrasonic pulse-echo method using an Olympus 5072 PR pulser/receiver. The sample size used for thermal diffusivity and sound velocity measurements was about $10 \times 10 \times 1.5$ mm³. The electrical and thermal properties of all the samples were measured in the same direction (perpendicular to the direction of pressure). The uncertainties of the measured Seebeck coefficient and electrical conductivity are approximately 3% and 5%, respectively. The uncertainty of thermal conductivity measurement is about 10%. As a result, the uncertainty of $ZT$ is estimated to be about 20%.

### Reporting summary

Further information on research design is available in the Nature Portfolio Reporting Summary linked to this article.

## Data availability

The authors declare that the data supporting the findings of this study are available within the paper and its Supplementary Information files. Source data are provided at https://doi.org/10.6084/m9.figshare.22560580.v1. Any other relevant data are also available upon reasonable request from Y.-H.L.

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

## Acknowledgements

This work was financially supported by the Basic Science Center Project of the National Natural Science Foundation of China under grant No. 52388201, the National Natural Science Foundation of China under grant No. 51772016, 52172211, 11934007, 11874194, the Science and Technology Innovation Committee Foundation of Shenzhen under grant No. JCYJ20200109141205978 and the Outstanding Talents Training Fund in Shenzhen. D.Y. acknowledges the beam time granted by ANSTO (P13964). B.W. thanks the Fundamental Research Funds for the Universities of Henan Province (No. NSFRF220421). Z.Z. acknowledges financial support from the Shuimu Tsinghua Scholar Program. We thank the SUSTech Core Research Facilities for use of the STEM instrument and we also thank the valuable advice from Dr. Bingbing Yang and Dr. Shun Lan.

## Author contributions

Z.Z. and Y.-H.L. proposed the research. Z.Z. carried out the experiments and most measurements. Z.Z. also wrote the manuscript. Y.H., D.H., and J.H. took part in the execution of the TEM experiments and analyses. D.Y. conducted the INS experiments and B.W. contributed related analyses. J.-L.L, Y.Y. and Y.Z. helped the analyses of electrical and thermal transport properties. C.-W.N. reviewed the manuscript and provided valuable advice. Y.-H.L., J.-L.L., B.W., Y.Y., Y.Z., W.Z., and M.Z. participated in the coordination of the study and revised the manuscript more than once. All authors read and approved the final manuscript.

## Competing interests

The authors declare no competing interests.
