## [Peer Review File · Nature Communications]

Compositing Effects for High Thermoelectric Performance of Cu₂Se-Based MaterialsREVIEWER COMMENTS

Reviewer #1 (Remarks to the Author):

The manuscript "Compositing Effects for High Thermoelectric Performance of Cu₂Se-Based Materials" presents improving thermoelectric performance through compositing of Cu₂Se, BiCuSeO based materials, and graphene. The work seems to be original, however, the manuscript, in my opinion, does not look adequate for publication in Nature Communications because of its relatively low potential significance for this journal. My detailed opinions are placed in bold as followings.

In the Abstract,

1) Here, we report a fast preparation method of self-propagating high-temperature synthesis to realize in-situ compositing of BiCuSeO and Cu₂Se to optimize the service stability.

: Similar preparation method was adopted to synthesize similar materials of BiCuSeO and Cu₂Se in the ref #34. The reference provides stability test results performed under high current density, which is good to check the stability of Cu₂Se. The manuscript presents a few cycle test only, without test under high current density.

2) Additionally, using the interface design by introducing graphene in these composites, the carrier mobility can be obviously enhanced

: This effect was already presented in references given by the authors (ref #42).

3), and the strong phonon scatterings can lead to lower lattice thermal conductivity

: This effect was already presented in references given by the authors. The manuscript does not show clear relationship between graphene amounts and transport properties, which is reported in references

4) Ultimately, the Cu₂Se-BiCuSeO-graphene composites presented excellent thermoelectric properties with a ZT_{max} value of ~2.82 at 1000 K and a ZT_{ave} value of ~1.73 at 473 K to 1000 K

: The efficacy of graphene is not obvious. Thermoelectric performance is not significantly enhanced compared to results in ref #34, which reports the peak figure of merit ZT value of ≈2.7 at 973 K and high average ZT value of ≈1.5 between 400 and 973 K for the Cu₂Se/BiCuSeO composites, without graphene. Considering precision of each measurement, electrical conductivity, thermal conductivity, and thermopower, the difference between 1.73 and 1.5 of average ZT does not seem to be meaningful, specially for materials with low thermal conductivity such as Cu₂Se based materials.

5) This work provides a facile and effective strategy to largely improve the performance of Cu₂Se-based thermoelectric materials, which could be furtherly adopted in other thermoelectric systems

: They simply combined 2 known strategies (Cu₂Se-BiCuSeO composite, graphene composites), and the effect of combination is not beneficial.

Reviewer #2 (Remarks to the Author):

The manuscript reports on the synthesis and in-situ compositing of BiCuSeO and Cu₂Se to optimize the service stability. Cu₂Se is known to be an excellent candidate for waste heat recovery, yet suffer from Cu diffusion leading to compositional instability under extended thermal load. Authors have developed a fast preparation method of self-propagating high-temperature synthesis to realize in-situ compositing of BiCuSeO and Cu₂ Se to optimize the service stability. In addition, by introducing graphene in these composites they demonstrated enhancement of carrier mobility, accompanied by a lower lattice thermal conductivity. The

resultant Cu₂Se-BiCuSeO-graphene composites presented excellent thermoelectric properties with a figure-of-merit (ZT_{max}) value of ~2.82 at 1000 K. Authors suggest that the facile and effective strategy to largely improve the performance of Cu₂Se-based thermoelectric materials, could be furtherly adopted in other thermoelectric systems.

There has been intensive research in the field of thermoelectrics to identify materials with promising figure-of-merit values at elevated temperatures, along with a long-term service stability. Cu₂Se is one of the promising thermoelectric (TE) materials for which ZT_{max} values over 2.0 has been reported. Authors have developed a synthetic strategy, combining in-situ composite formation and interface engineering with graphene, making it possible to synergistically optimize electrical and thermal properties for a favorable ZT. This is a striking achievement in the progress recorded for this material so far, and the suggested methodology can be used for improving the efficiency of other TE materials.

The manuscript is well organized, and a thorough experimental evaluation was undertaken to unveil the electronic and thermal transport characteristics. There are many small typos, which should be sorted by a language revision. There are few points listed below though, if explained in a bit more details would improve the clarity and reproducibility of the methods by other researchers -before the manuscript should be accepted.

In the experimental details, in the Methods section, the authors describe the Sample preparation very roughly (powders were thoroughly mixed by hand grinding, then cold pressed into pellets and underwent the SHS process.). What are the process details for the SHS process? There is no mention to the mechanism of triggering the self-propagation reaction, or the critical energy/temperature threshold that needs to be attained to initiate the SHS process.

- What is typical batch size, and if there are any limitations to that as experimentally experienced?

- How is the mixing with graphene performed? Manually by grinding, or by using a ball mill system? As the homogeneity of graphene in the matrix is essential to achieve the obtained results, the way of preparing the sample should be as detailed as possible.

Phase identification of the composites, presented in Fig S4, shows peaks of Bi₂O₃, that the authors suggest these are minor peaks and reveals that Bi₂O₃ “formed different scales of precipitates embedded in the matrix”. Are these analyses done on powder samples, or after the SPS process? If after sintering, what is the orientation of the measured sample/pellet with respect to the sintering/pressure direction. A sketch would be useful as inset to the XRD figure. Furthermore, as the process of Cu₂Se-BPCCSO composite formation is a single step SHS process, there must be some other impurity phases -besides Bi₂O₃, based on the stoichiometry of the initial mixture. Naturally, those phases should also play a role in the matrix, perhaps being scattering centers. Did the authors identify other phases in HAADF imaging, if not in the XRD? How do the authors judge the success of the SHS process, considering the presence of the impurity phases?

All the transport data are presented in clear form, while the density of SPS'd samples are presented in a table in the supplementary file. It would be fruitful to make an explicit mention to the percentage of densification -or packing density (e.g. >90% theoretical density) along with the measured density values. Information about sample geometry and dimensions used for various transport and mechanical testing should be given in the relevant sections. The PF has been measured in perpendicular plane to the pressing direction, as specified in the experimental methods. How about the measurement of thermal diffusivity and sound velocity? Are they measured in the same plane or perpendicular to it?

The estimation of Cp for Cu₂Se is performed using DSC, while that for BPCCSO, Cu₂Se-BPCCSO composites, and 0.9Cu₂Se-0.1BPCCSO-x wt% graphene composites was

estimated according to the Neumann-Kopp rule. This rule is known to give good predictions at room-temperature conditions, but poorly at elevated temperatures. Has there been any experimental control of reliability of estimated C_p at elevated temperatures for the composite materials? If so, this would be expected to change the magnitude of the thermal conductivity values presented. Could authors comment on the use of Neumann-Kopp rule?

Assessed from the Seebeck coefficient Cu_2Se is a p-type semiconductor, as well as the composites with BPCCO and graphene. Is the graphene used in this work n- or p-type? Would the type of graphene make a difference in the observed effect of increased mobility?

A last question is on the versatility of the developed methodology to other materials systems. Can this strategy be used to develop an n-type counterpart (Ag-Se, or Ag doped version of Cu_2Se) stabilized using the same strategy?

Response to Reviewers

Reviewer #1:

The manuscript "Compositing Effects for High Thermoelectric Performance of Cu₂Se-Based Materials" presents improving thermoelectric performance through compositing of Cu₂Se, BiCuSeO based materials, and graphene. The work seems to be original, however, the manuscript, in my opinion, does not look adequate for publication in Nature Communications because of its relatively low potential significance for this journal.

Response: Thanks for the referee's comments. In this work, we developed a facile and effective strategy to largely improve the performance of Cu₂Se-based thermoelectric materials, which could be furtherly adopted in other thermoelectric systems. And a record high ZT_{\max} value of ~2.82 at 1000 K and ZT_{ave} value of ~1.73 at 473 K to 1000 K was realized, which exhibits good potential for further practical applications. Furthermore, we deeply unveiled the physical novelty by systematically studying the atomic structure, carrier transport properties, phonon dispersion, and thermoelectric properties.

After carefully reading the comments, we realize that the major merits of our work have been recognized by Reviewer #2: "The resultant Cu₂Se-BiCuSeO-graphene composites presented excellent thermoelectric properties..... Authors have developed a synthetic strategy..... making it possible to synergistically optimize electrical and thermal properties for a favorable ZT . This is a striking achievement in the progress

recorded for this material so far, and the suggested methodology can be used for improving the efficiency of other thermoelectric materials."

Therefore, we believe that the study presented in the manuscript should be interesting for the thermoelectric community.

As per the referee's valuable comments, we have added the related discussions to improve this manuscript and make conclusive enough of this work for publication.

1. "Here, we report a fast preparation method of self-propagating high-temperature synthesis to realize in-situ compositing of BiCuSeO and Cu₂Se to optimize the service stability." Similar preparation method was adopted to synthesize similar materials of BiCuSeO and Cu₂Se in the ref #34. The reference provides stability test results performed under high current density, which is good to check the stability of Cu₂Se. The manuscript presents a few cycle tests only, without test under high current density.

Response: Thanks for the referee's comments. Actually, there are big differences in the preparation method and materials between this work and the previous work in Ref #34. In this work, the Cu₂Se-BiCuSeO composites were in-situ formed by a one-step SHS process directly from the raw materials. However, they first synthesized Bi₂O₂Se by thermal explosion method and then conducted the SHS process (*Adv. Mater.*, 2020, 32, 2003730). Herein, our one-step method is much more time- and cost-effective, which could be more easily extended to use in other thermoelectric systems. In addition, the materials we designed are totally different from the ones in Ref #34. What we

investigated was $\text{Cu}_2\text{Se-BiCuSeO}$ composites (the mole fraction of BiCuSeO was 10%-30%), while the mole fraction of BiCuSeO in Ref #34 was no more than 0.5%, which could even hardly be detected. That is to say, their materials should still be Cu_2Se essentially instead of $\text{Cu}_2\text{Se-BiCuSeO}$ composites in Ref #34. Our work indicates more BiCuSeO could bring largely improved bending strength for Cu_2Se , which is meaningful for future practical applications (**Figure 1f**). Furthermore, our work, for the first time, introduced graphene into $\text{Cu}_2\text{Se-BiCuSeO}$ composites, and the $\text{Cu}_2\text{Se-BiCuSeO-graphene}$ composites have never been reported in previous works. Thus, the materials we prepared are not similar to the ones in Ref #34.

As for the service stability, the heating and cooling cycle test was usually used to test the service stability of thermoelectric materials (*Science*, 2022, 375, 1385; *Science*, 2022, 377, 208; *Nat. Commun.*, 2021, 12, 3550; *Energy Environ. Sci.*, 2017, 10, 1928). Most of them were only under once or twice cycling tests. Here, the three times heating and cooling cycles could reflect good service stability to some extent, and the stability under high current density should also be good based on the results in Ref #34. We have added more discussions on service stability in the revised manuscript (Please see Lines 5-9, Page 7). For Cu_2Se -based materials, there was a thermodynamic threshold for stability, which was named critical voltage. Only when the voltage applied on Cu_2Se exceeds the critical voltage, Cu deposition occurs on the cathode yielding poor stability. However, when the voltage is below the critical voltage, the mobile Cu ions just form a steady concentration gradient from the anode to the cathode without depositing from the material (*Nat. Commun.*, 2018, 9, 2910). In this case, Cu_2Se has

good stability as other state-of-the-art thermoelectric materials. Recently, it is proposed that electrical conductivity has a positive correlation with critical voltage and can be a simple indicator to rapidly predict the stability of Cu₂Se-based materials, where the higher electrical conductivity is, the better stability will be (*Mater. Today Phys.*, 2021, 21, 100550). Compared to the sample 0.999Cu₂Se_{1.005}-0.001BiCuSeO with good stability which has been proven in Ref #34, samples such as 0.9Cu₂Se-0.1BPCCSO-0.035 wt% graphene in this work showed higher electrical conductivity, indicating better stability. The comparison on the electrical conductivity of samples is shown in

Figure R1.

Figure R1 Comparison on electrical conductivity (σ) of samples in this work and Ref #34

2. “Additionally, using the interface design by introducing graphene in these composites, the carrier mobility can be obviously enhanced.” This effect was already presented in references given by the authors (ref #42).

Response: Thanks for the referee's comments. For the Cu₂Se-BiCuSeO composites, the relatively low carrier mobility hinders the further improvement of thermoelectric performance, and our work is the first time to introduce graphene for optimizing their carrier mobility. This strategy brought a ~50% improvement in carrier mobility, and the highest carrier mobility of ~40 cm² V⁻¹ s⁻¹ was much higher than those of the current state-of-the-art Cu₂Se-based materials (e.g. *Energy Environ. Sci.*, 2017, 10, 1928; *InfoMat*, 2019, 1, 108; *Adv. Mater.*, 2020, 32, 2003730; *Adv. Funct. Mater.*, 2020, 30, 1908315). The comparisons on carrier mobility have been presented in **Figure 3c**. In addition, differently, Cu₂Se-BiCuSeO composites possessed p-type conducting behaviors, while Bi₂O₂Se was an n-type semiconductor in Ref #42. Although there were Cu₂Se composited with carbon materials reported in the literature, they were mainly focused on the thermal properties (*Energy Environ. Sci.*, 2020, 13, 3307). Here, our work indicates that the appropriate content of graphene can be used to optimize the carrier mobility of Cu₂Se, which will enlighten the readers for future research.

3. "and the strong phonon scatterings can lead to lower lattice thermal conductivity." This effect was already presented in references given by the authors. The manuscript does not show clear relationship between graphene amounts and transport properties, which is reported in references.

Response: Thanks for the referee's comments. In this work, we not only presented the phenomenon of lower lattice thermal conductivity, but we used the inelastic neutron scattering (INS) techniques and sound velocity test to analyze the phonon transport

properties in such complex composites. We quantitatively described the overlapping of phonon modes by calculating the overlapping factor and provided a good explanation for the ultralow thermal conductivity of Cu₂Se-BPCCSO-graphene composites, which is a progress compared with other previous works.

Furthermore, as per the referee's suggestion, we have added more related discussion about the relationship between graphene amounts and transport properties. The added corresponding sentences in the revised manuscript are as follows.

Pages 10, 12. – “Charge transport properties” section:

-With increasing the contents of graphene, the electrical conductivity first increased and then decreased.

- It is also noted that when the content of graphene reached 0.04 wt%, the hall mobility decreased sharply, which caused the lower electrical conductivity of 0.9Cu₂Se-0.1BPCCSO-0.04 wt% graphene composite.

- The change trend of the Seebeck coefficient for samples with different graphene contents was related to the electrical conductivity, but the variety was small, especially at room temperature.

Pages 13, 14. – “Thermal transport properties” section:

- With the increasing contents of graphene from x=0.02 to x=0.04, the thermal conductivity first increased and then decreased.

- With the increasing content of graphene, the lattice thermal conductivity increased, which may be ascribed to the aggregation of graphene that reduced the interfacial phonon scattering effects.

4. “Ultimately, the $\text{Cu}_2\text{Se-BiCuSeO}$ -graphene composites presented excellent thermoelectric properties with a ZT_{max} value of ~ 2.82 at 1000 K and a ZT_{ave} value of ~ 1.73 at 473 K to 1000 K.” The efficacy of graphene is not obvious. Thermoelectric performance is not significantly enhanced compared to results in ref #34, which reports the peak figure of merit ZT value of ≈ 2.7 at 973 K and high average ZT value of ≈ 1.5 between 400 and 973 K for the $\text{Cu}_2\text{Se/BiCuSeO}$ composites, without graphene. Considering the precision of each measurement, electrical conductivity, thermal conductivity, and thermopower, the difference between 1.73 and 1.5 of average ZT does not seem to be meaningful, especially for materials with low thermal conductivity such as Cu_2Se based materials.

Response: Thanks for the referee’s comments. Compared to $0.9\text{Cu}_2\text{Se-0.1BPCCSO}$, the ZT_{max} value increased by $\sim 32\%$, which indicates the significant effects on improving thermoelectric performance by introducing graphene. Also, it is actually very hard to further improve the average ZT values for thermoelectric materials with high performance. For instance, the average ZT values can only be increased from 0.86 to 0.90 and from 1.51 to 1.56 for PbSe- and GeTe- based thermoelectric materials, respectively (*Nat. Commun.*, 2022, 13, 6449; *Nat. Commun.*, 2022, 13, 6087). Thus, the increase in the average ZT value from 1.50 to 1.73 achieved in this work should be meaningful. Furthermore, when compared to the sample $0.999\text{Cu}_2\text{Se}_{1.005}\text{-0.001BiCuSeO}$ with the best thermoelectric performance in Ref #34, sample $0.9\text{Cu}_2\text{Se-0.1BPCCSO-0.035 wt\% graphene}$ in this work not only had higher peak ZT and average ZT values, but possessed much larger average power factor of $14.49 \mu\text{W}$

$\text{cm}^{-1} \text{K}^{-2}$. The average power factor reached in this work was ~33% larger than that of $0.999\text{Cu}_2\text{Se}_{1.005}\text{-}0.001\text{BiCuSeO}$ in Ref #34. For materials with low thermal conductivity, the significant improvement in electric performance should be more meaningful. **Table R1** shows the comparison on thermoelectric performance including peak ZT (ZT_{max}), average ZT (ZT_{ave}), average power factor (PF_{ave}), and Hall mobility (μ_{H}) of state-of-the-art Cu_2Se -based systems from this work and references. Herein, the thermoelectric performance achieved in this work is outstanding.

Table R1 Comparison on the thermoelectric performance of Cu_2Se -based systems

Composition	ZT_{max}	ZT_{ave}	PF_{ave} [$\mu\text{W cm}^{-1} \text{K}^{-2}$]	μ_{H} [$\text{cm}^2 \text{V}^{-1} \text{s}^{-1}$]	Ref.
Cu_2Se - 0.10 wt% carbon/coated boron	2.23 @1000 K	1.49	8.44	/	InfoMat , 2019, 1, 108
Cu_2Se - 0.75 wt% CNTs	2.40 @1000 K	1.44	8.80	15.0	Energy Environ. Sci. , 2017, 10, 1928
$0.999\text{Cu}_2\text{Se}_{1.005}\text{-}$ 0.001BiCuSeO	2.67 @973 K	1.50	10.86	13.7	Adv. Mater. , 2020, 32, 2003730
$\text{Cu}_2\text{Se}_{0.92}\text{S}_{0.08}$	2.0 @1000 K	1.19	10.60	17.5	Chem. Mater. , 2017, 29, 6367
$\text{Cu}_{1.96}\text{Se}_{0.8}\text{S}_{0.2}$	2.0 @1000 K	1.11	10.37	10.7	Adv. Funct. Mater. , 2020, 30, 1908315
0.9Cu_2Se- 0.1BPCCSO-0.035 wt% Graphene	2.82 @1000 K	1.73	14.49	38.1	This work

5. “This work provides a facile and effective strategy to largely improve the performance of Cu_2Se -based thermoelectric materials, which could be furtherly adopted in other thermoelectric systems.” They simply combined 2 known strategies (Cu_2Se - BiCuSeO composite, graphene composites), and the effect of combination is not beneficial.

Response: Thanks for the referee’s comments. In this work, the facile and effective

strategy of combining in-situ composite formation and interface engineering was first used in Cu_2Se -based materials, which could be furtherly extended to use in other thermoelectric systems. Also, we successfully realized a synergistic optimization of service stability and thermoelectric performance by this strategy, which indicates the effect of the strategy is beneficial. Because of the aforementioned novelties which are strongly supported by our experimental data and theoretical analyses, we do believe that our research constitutes a breakthrough and the manuscript is of very high quality. Therefore, we think that our manuscript should be interesting for the thermoelectric community and show high potential significance for this journal.

Reviewer #2:

The manuscript reports on the synthesis and in-situ compositing of BiCuSeO and Cu_2Se to optimize the service stability. Cu_2Se is known to be an excellent candidate for waste heat recovery, yet suffer from Cu diffusion leading to compositional instability under extended thermal load. Authors have developed a fast preparation method of self-propagating high-temperature synthesis to realize in-situ compositing of BiCuSeO and Cu_2Se to optimize the service stability. In addition, by introducing graphene in these composites they demonstrated enhancement of carrier mobility, accompanied by a lower lattice thermal conductivity. The resultant Cu_2Se - BiCuSeO -graphene composites presented excellent thermoelectric properties with a figure-of-merit (ZT_{max}) value of ~ 2.82 at 1000 K. Authors suggest that the facile and effective strategy to

largely improve the performance of Cu₂Se-based thermoelectric materials, could be furtherly adopted in other thermoelectric systems.

There has been intensive research in the field of thermoelectrics to identify materials with promising figure-of-merit values at elevated temperatures, along with a long-term service stability. Cu₂Se is one of the promising thermoelectric (TE) materials for which ZT_{\max} values over 2.0 has been reported. Authors have developed a synthetic strategy, combining in-situ composite formation and interface engineering with graphene, making it possible to synergistically optimize electrical and thermal properties for a favorable ZT . This is a striking achievement in the progress recorded for this material so far, and the suggested methodology can be used for improving the efficiency of other TE materials.

The manuscript is well organized, and a thorough experimental evaluation was undertaken to unveil the electronic and thermal transport characteristics. There are many small typos, which should be sorted by a language revision. There are few points listed below though, if explained in a bit more details would improve the clarity and reproducibility of the methods by other researchers -before the manuscript should be accepted.

Response: Thanks very much for the referee's comments on our work being "a striking achievement in the progress recorded for this material". For the typos, we have carefully checked and revised our manuscript.

1. In the experimental details, in the Methods section, the authors describe the sample

preparation very roughly (powders were thoroughly mixed by hand grinding, then cold pressed into pellets and underwent the SHS process.). What are the process details for the SHS process? There is no mention to the mechanism of triggering the self-propagation reaction, or the critical energy/temperature threshold that needs to be attained to initiate the SHS process.

- What is typical batch size, and if there are any limitations to that as experimentally experienced?

- How is the mixing with graphene performed? Manually by grinding, or by using a ball mill system? As the homogeneity of graphene in the matrix is essential to achieve the obtained results, the way of preparing the sample should be as detailed as possible.

Response: As per the referee's suggestion, we have added the experimental details in the revised manuscript. To initiate the SHS process, the adiabatic temperature that represents the maximum temperature to which the reacting compact is raised as the combustion wave passes through, must be high enough to melt the lower melting point component (*Nat. Commun.* 2014, 5, 4908). In this work, the lowest melting point from Se is around 494 K. Thus, the alcohol lamp was used as a heating source. For each sample, the weight of the cold-pressed pellet was about 10 g. The size of the batch is not limited which mainly depends on the size of the container. The graphene was thoroughly mixed with obtained powders by hand grinding for more than 30 min. The added corresponding sentences in the revised manuscript are as follows.

Pages 24, 25. – “Sample preparation” section:

For each sample, the weight of the pellet was about 10 g. The pellet was put in an alumina crucible. The SHS process started by heating the bottom of the crucible to the ignition temperature with an alcohol lamp. Once the reaction began from the lower melting point component, we put the lid on the crucible and moved the alcohol lamp away. The combustion wave was persisted by the energy released from the initial reaction and spread to the whole pellet in few seconds. The obtained bulks were ground into fine powders and fully mixed with graphene (XFNANO, 0.5-5 μm in diameter, 0.8 nm in thickness) by hand grinding for more than 30 min.

2. Phase identification of the composites, presented in Fig S4, shows peaks of Bi_2O_3 , that the authors suggest these are minor peaks and reveals that Bi_2O_3 “formed different scales of precipitates embedded in the matrix”. Are these analyses done on powder samples, or after the SPS process? If after sintering, what is the orientation of the measured sample/pellet with respect to the sintering/pressure direction. A sketch would be useful as inset to the XRD figure. Furthermore, as the process of Cu_2Se -BPCCSO composite formation is a single step SHS process, there must be some other impurity phases -besides Bi_2O_3 , based on the stoichiometry of the initial mixture. Naturally, those phases should also play a role in the matrix, perhaps being scattering centers. Did the authors identify other phases in HAADF imaging, if not in the XRD? How do the authors judge the success of the SHS process, considering the presence of the impurity phases?

Response: Thanks for the referee's comments. The samples used for XRD measurement were fine powders crushed and ground from the sintered bulks, which has been explained in the revised manuscript. Since the XRD detection was under a very slow scanning rate of 1° min^{-1} , the impurity phases should be observed if the contents were above the detected limitation. Besides, there were no other impurity phases detected by STEM-HAADF imaging. Thus, we think there were no other impurity phases or there were few impurity phases that could not be detected, and the effects on thermoelectric performance could be neglected. In addition, as the SHS process was conducted in air, a small amount of Bi inevitably evaporated, and then parts of Bi_2O_3 were unreacted as impurity phases according to the chemical reaction ($\text{Bi} + \text{Bi}_2\text{O}_3 + 3\text{Cu} + 3\text{Se} \rightarrow 3\text{BiCuSeO}$). In this case, to judge the success of the SHS process, we focused more on the formation of the expected major phases. The added corresponding sentences in the revised manuscript are as follows.

Page 25. – “Structural characterization” section:

The samples for XRD measurement were fine powders crushed and ground from the sintered bulks.

Page 8. – “Structure of Cu_2Se -BPCCSO-graphene” section:

The existence of Bi_2O_3 may be ascribed to the evaporation of Bi elements, and a small amount of Bi_2O_3 was unreacted as impurity phases.

3. All the transport data are presented in clear form, while the density of SPS'd samples is presented in a table in the supplementary file. It would be fruitful to make an explicit mention to the percentage of densification -or packing density (e.g. >90% theoretical

density) along with the measured density values. Information about sample geometry and dimensions used for various transport and mechanical testing should be given in the relevant sections. The PF has been measured in perpendicular plane to the pressing direction, as specified in the experimental methods. How about the measurement of thermal diffusivity and sound velocity? Are they measured in the same plane or perpendicular to it?

Response: As per the referee's suggestion, we have made an explicit mention to the percentage of densification and added the information about sample geometry and dimensions used for property measurement in the methods part. The thermal diffusivity and sound velocity were also measured in perpendicular plane to the pressing direction, which has been explained in the revised manuscript. The added corresponding sentences in the revised manuscript are as follows.

Page 5. – “Cu₂Se-BPCCSO composites” section:

All the samples were highly dense with a relative density of over 95%, as shown in Supplementary Table 1.

Pages 26, 27. – “Transport property measurement” section:

-The sample size was about 3×3×11 mm³ (For measuring electrical conductivity and Seebeck coefficient).

-The sample size was about 10×10×0.5 mm³ (For Hall measurement).

-The sample size used for thermal diffusivity and sound velocity measurements was about 10×10×1.5 mm³.

-The electrical and thermal properties of all the samples were measured in the same

direction (perpendicular to the direction of pressure).

4. The estimation of C_p for Cu_2Se is performed using DSC, while that for BPCCSO, Cu_2Se -BPCCSO composites, and $0.9\text{Cu}_2\text{Se}$ - 0.1BPCCSO - x wt% graphene composites was estimated according to the Neumann-Kopp rule. This rule is known to give good predictions at room-temperature conditions, but poorly at elevated temperatures. Has there been any experimental control of reliability of estimated C_p at elevated temperatures for the composite materials? If so, this would be expected to change the magnitude of the thermal conductivity values presented. Could authors comment on the use of Neumann-Kopp rule?

Response: Thanks for the referee's comments. The Neumann-Kopp rule is the simplest and highly universal method for estimating the heat capacity of compounds (*Thermochim. Acta*, 2002, 395, 27; *Phys. Rev. B*, 2006, 74, 144109), which has also been widely used in thermoelectric composite materials (*Mater. Today Phys.*, 2022, 27, 100808; *Adv. Mater.*, 2022, 34, 2109952; *ACS Appl. Mater. Interfaces*, 2022, 14, 55780; *Acta Mater.*, 2023, 244, 118564). Considering Cu_2Se is a phase-transition material with different heat capacity values at each stage, we measured the temperature-dependent heat capacity of Cu_2Se by using DSC. The measured values were close to the values from the literature (*Energy Environ. Sci.*, 2017, 10, 1928; *Nano Energy*, 2018, 53, 993). Based on the measured temperature-dependent heat capacity values of Cu_2Se , we estimated the heat capacity values of composites at each temperature by using the Neumann-Kopp rule. The estimated values for the composites should be relatively

reliable. Considering the difficulties in accurately measuring C_p by using DSC, we used the aforementioned methods to estimate the C_p values of composites, which is more beneficial for comparing the thermal properties and unveiling the underlying mechanisms in phonon transport.

5. Assessed from the Seebeck coefficient Cu_2Se is a p-type semiconductor, as well as the composites with BPCCO and graphene. Is the graphene used in this work n- or p-type? Would the type of graphene make a difference in the observed effect of increased mobility?

Response: It is a good question. The graphene used in this work was n-type, which could be confirmed by the decreasing trend of carrier concentration (Fig. 3b), and the corresponding sentence has been added in the revised manuscript (Please see Line 21, Page 10). Since the contents of graphene were very low, the addition of graphene had only a small effect on the carrier concentration but largely improved carrier mobility. For the n-type thermoelectric materials such as $\text{Bi}_2\text{O}_2\text{Se}$, the introduced graphite nanosheets simultaneously improved the carrier concentration and carrier mobility (*Adv. Funct. Mater.*, 2022, 32, 2202927). Thus, the type of graphene should barely affect the observed effect of increased mobility. Related research could be developed in the future.

6. A last question is on the versatility of the developed methodology to other materials systems. Can this strategy be used to develop an n-type counterpart (Ag-Se, or Ag doped version of Cu_2Se) stabilized using the same strategy?

Response: It is a good point. For the n-type superionic materials like Ag_2Se or Ag_2Te , in our opinion, this strategy is expected to be adopted. Since Ag_2Se or Ag_2Te could also be synthesized by SHS process (*ACS Appl. Mater. Interface*, 2022, 14, 5439; *Mater. Trans.*, 2020, 61, 2402; *AIP Adv.*, 2023, 13, 015016), n-type promising oxygen-containing materials $\text{Bi}_2\text{O}_2\text{Se}$ or $\text{Bi}_2\text{O}_2\text{Te}$ could be in-situ composited with them to improve their service stability. Then, we could introduce graphene or other carbon materials to further optimize their thermoelectric performance. The related work could be developed in the future. At least, this work laid a solid foundation for the following research.

REVIEWERS' COMMENTS

Reviewer #2 (Remarks to the Author):

The authors have responded my queries in a detailed and convincing manner. I believe that the manuscript can be accepted in the revised form.

Response to Reviewers

Reviewer #2:

The authors have responded my queries in a detailed and convincing manner. I believe that the manuscript can be accepted in the revised form.

Response: Thanks for the referee's positive evaluation and valuable suggestions on our manuscript.